# Optimization of Dissolved Silica Removal from Reverse Osmosis Concentrate by *Gedaniella flavovirens* for Enhanced Water Recovery

**Han Gao** [1], **Shinya Sato** [2] , **Hitoshi Kodamatani** [3] , **Takahiro Fujioka** [4] , **Kenneth P. Ishida** [5]
and **Keisuke Ikehata** [1,*]

1    Ingram School of Engineering, Texas State University, San Marcos, TX 78666, USA; keeehangao@gmail.com
2    Department of Marine Science and Technology, Fukui Prefectural University, Fukui 917-0003, Japan;
     ssato@fpu.ac.jp
3    Graduate School of Science and Engineering, Kagoshima University, Kagoshima 890-8544, Japan;
     kodama@sci.kagoshima-u.ac.jp
4    Graduate School of Engineering, Nagasaki University, Nagasaki 852-8521, Japan; tfujioka@nagasaki-u.ac.jp
5    Orange County Water District, Fountain Valley, CA 92708, USA; kenishida714@gmail.com
*    Correspondence: kikehata@txstate.edu; Tel.: +1-512-245-0855

**Abstract:** Photobiological treatment of reverse osmosis concentrate (ROC) using brackish diatoms is a green and sustainable technology that can enhance water recovery by removing dissolved silica from ROC while producing beneficial biomass. This study aimed to determine the optimum conditions for the photobiological treatment of ROC obtained from a full-scale advanced water purification facility using *Gedaniella flavovirens* Psetr3. While light color presented minor impacts on the silica uptake rate, the impact of color intensity was significant. The uptake rate improved from $28 \pm 1$ to $48 \pm 7$ mg/L/day by increasing photosynthetically active radiation (PAR) from 50 to 310 µmol m$^{-2}$ s$^{-1}$. Increasing the PAR further did not improve the performance. The optimum temperature was around 23–30 °C. While the silica uptake was slower at 10 °C, *G. flavovirens* Psetr3 was unable to survive at 40 °C. Experiments using sunlight as a light source verified the impact of temperature on the silica uptake and the detrimental effect of ultraviolet radiation on this diatom. The sunlight-based treatment effectively removed *N*-nitrosodimethylamine. The results of this study are being used in subsequent pilot-scale investigations and full-scale technoeconomic analysis and will contribute to the further development of this sustainable water technology.

**Keywords:** concentrate management; desalination; diatoms; green technology; nitrosamines; photobiological treatment; potable reuse; reverse osmosis; sustainable water resources

## 1. Introduction

Advanced purification and reuse of secondary- or tertiary-treated municipal wastewater is an increasingly popular approach to expand the potable water supply portfolio of utilities in populous urban cities in arid and semi-arid areas such as California, Arizona, and Texas [1–3]. Domestic wastewater is an important, locally available, sustainable water resource that is still largely unutilized. Reverse osmosis (RO) is a critical technology in advanced water purification facilities (AWPFs), along with microfiltration (MF) or ultrafiltration (UF) and the ultraviolet/hydrogen peroxide advanced oxidation process (UV/H$_2$O$_2$ AOP). In the treatment of municipal wastewater, the RO process is utilized as an effective barrier for removing dissolved organic and inorganic constituents, as well as microbial constituents such as viruses, bacteria, and protozoa [4–7]. Where dissolved solids (i.e., salinity) removal is crucial, RO is essential because other unit processes, including MF, UF, UV AOP, and ozone–biologically active carbon filtration cannot achieve this task. While the RO process can produce high-quality purified water for potable purposes, it generates

a waste stream, called concentrate or brine, that needs to be properly disposed into an aquatic environment [8,9]. Current RO concentrate (ROC) management methods at AWPFs include discharge to ocean, surface water or sewer, deep well injection, and evaporation. However, additional fresh water could be recovered from the ROC because its salinity is generally less than 10,000 mg/L as total dissolved solids (TDS) [10]. Further treatment of ROC (e.g., secondary RO) can increase water recovery and reduce concentrate volume for disposal. The limiting factor for the water recovery is the formation of inorganic scalants such as calcium carbonate, calcium phosphate, barium sulfate, and silica on the membrane surface [11–15]. In addition, nutrients and other organic and inorganic contaminants in AWPF ROC can pose a threat to the environments of receiving water bodies [8,16,17].

A new photobiological treatment process using brackish diatoms for scalant and nutrient removal has been developed for the treatment of ROC from AWPFs and brackish groundwater desalination facilities to enhance water recovery, as illustrated in Figure 1 [18]. Brackish diatoms such as *Gedaniella flavovirens* (formerly named *Pseudostaurosira trainorii*) [19], *Nitzschia* spp., *Anomoeoneis* spp., and *Halamphora* spp. were found to be useful in the scalant and nutrient removal from ROC [20]. The feasibility of additional water recovery from the photobiologically treated ROC has been demonstrated by laboratory-scale secondary RO experiments and pilot-scale studies [18]. When ROC samples from six different AWPFs were compared for treatability, a high concentration of ammonia-N (>16 mg/L) was found to be toxic to *G. flavovirens* and inhibitory to dissolved silica removal [10]. Light sources including commercial compact fluorescent lamps, light-emitting diode (LED) bulbs, and natural sunlight have been tested and found to be useful in this process [18]. Further, the removal of trace organic contaminants such as atenolol, propranolol, and *N*-nitrosodimethylamine (NDMA) by the photobiological process has been observed [18].

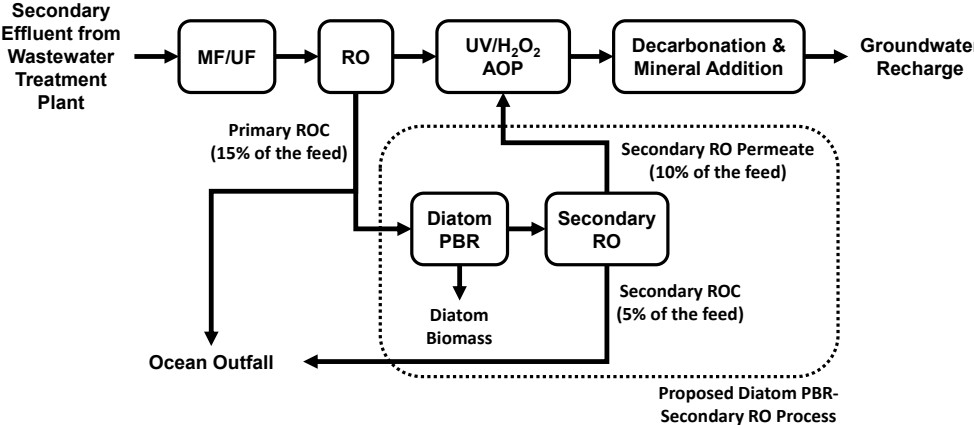

**Figure 1.** Simplified scheme of the diatom photobioreactor (PBR)–secondary reverse osmosis (RO) process for enhanced water recovery at a potable reuse facility. Abbreviations: MF = microfiltration, UF = ultrafiltration, $UV/H_2O_2$ AOP = ultraviolet/hydrogen peroxide advanced oxidation process.

Although previous research has highlighted the potential of this innovative green technology using a diatom photobioreactor (PBR) followed by secondary RO, there remains a gap in understanding the impact of light-related parameters (such as intensity, color, and duration) and temperature on dissolved silica. These operational parameters play a critical role in optimizing the growth of photosynthetic microalgae, including diatoms [21–23], and in the efficient operations of PBR systems [24–27]. Therefore, this study aimed to systematically investigate the optimum treatment conditions concerning light and temperature. Identifying these optimal conditions is crucial for subsequent research and development activities, including continuous flow pilot testing and technoeconomic analysis. Additionally, this study also evaluated the removal of NDMA from AWPF ROC via the photobiological process. This assessment is important given the regulatory limits on NDMA, particularly for certain AWPFs involved in indirect and direct potable water reuse [28–30].

## 2. Materials and Methods

### 2.1. ROC Samples

Three batches of ROC samples were collected at the end of the third stage of the RO process at the Orange County Water District (OCWD) Groundwater Replenishment System (GWRS) AWPF (Fountain Valley, CA, USA) and shipped to Texas State University (San Marcos, TX, USA). These ROC samples are referred to as GWRS ROC hereafter. The GWRS ROC sample collection dates were 23 September 2019, 11 September 2020, and 25 March 2021. The RO permeate recovery was 85%. The samples were kept refrigerated at 4 °C in the laboratory until use. Table 1 shows the average water quality of the GWRS ROC samples. No chloramine residuals were detected (<0.02 mg/L) at the time the samples were used in this study. GWRS ROC was used after microfiltration through either 0.8/0.2 μm syringe filters (Acrodisc® PF with Supor® membrane (hydrophilic polyethersulfone), sterile, 32 mm diameter, Pall Corporation, Port Washington, NY, USA) or 0.45 μm Pall Versapor® acrylic co-polymer membrane filters (45 mm diameter) with a vacuum filtration apparatus immediately before use and analysis.

**Table 1.** Average water quality of GWRS ROC samples used.

| Parameter | Average ± Standard Deviation |
|---|---|
| Sodium (mg/L) | 1200 ± 100 |
| Calcium (mg/L) | 779 ± 25 |
| Magnesium (mg/L) | 140 ± 10 |
| Iron (mg/L) | 0.3 ± 0.1 |
| Ammonia-N (mg/L) | 6.0 ± 0.8 |
| Chloride (mg/L) | 1670 ± 20 |
| Sulfate (mg/L) | 1000 ± 50 |
| Bicarbonate (mg/L) | 590 ± 90 |
| Nitrate-N (mg/L) | 60 ± 4 |
| Reactive silica (mg/L) | 131 ± 3 |
| Orthophosphate (mg/L) | 10.4 ± 1.6 |
| Total dissolved solids (mg/L) | 5380 ± 110 |
| Conductivity (mS/cm) | 8.03 ± 0.16 |
| Alkalinity (mg/L as $CaCO_3$) | 968 ± 148 |
| Chemical oxygen demand (mg/L) | 129 ± 7 |
| pH | 8.5 ± 0.4 |
| Apparent color at 455 nm (PtCo unit) | 145 ± 3 |

### 2.2. Diatom

A unialgal culture of brackish diatom *G. flavovirens* Psetr3 isolated from the bottom sands of Obuchi-numa Lake in Aomori Prefecture, Japan [20] was used in this study. Primary cultures of *G. flavovirens* were maintained in approximately 10 mL of filtered GWRS ROC in 15 mL clear polypropylene centrifuge tubes (SuperClear™, VWR International, Radnor, PA, USA) under continuous light at room temperature (22 ± 1 °C). Subcultures were created from the primary culture and were grown in approximately 40 mL of filtered GWRS ROC in VWR SuperClear™ 50 mL clear polypropylene centrifuge tubes along with primary cultures. The GWRS ROC in both primary and subcultures was replaced once a week. Subcultures were used as seed cultures for the bench-scale photobiological treatment experiments, typically after four weeks of continuous culturing.

### 2.3. Light and Temperature Measurements

The photosynthetically active radiation (PAR) at the positions where photobiological treatment vessels were placed was measured prior to and during the experiments using an Apogee Full Spectrum Quantum Meter (MQ-500, Logan, UT, USA). An Apogee MU-200 UV Meter (250–400 nm) and an SS-110 Visible Spectroradiometer (340–820 nm) were used to measure UV and visible radiation, respectively. Temperature was monitored by a USB

Temp Data Logger (EL-USB-1, Lascar Electronics, Erie, PA, USA) placed adjacent to the vessels during the experiment.

### 2.4. Photobiological Treatment—Indoor Experiments

A series of semi-batch photobiological treatment experiments were conducted in the laboratory to investigate the impact of light sources and temperature in a controlled environment. Clear 100 mL polystyrene coliform bottles without sodium thiosulfate preservative (Grainger, Lake Forest, IL, USA) were used as reaction vessels (Figure 2). Filtered GWRS ROC was transferred into the vessels, and a seed culture of *G. flavovirens* Psetr3 (0.19 ± 0.01 g/L) was added. Aliquots of the seed culture were collected in 1.5 mL microcentrifuge tubes to determine dry biomass weight. The vessels were capped tightly and placed in a five-gallon plastic bucket lined with a silver reflective bubble wrap sheet (S-11476, ULINE, Pleasant Prairie, WI, USA) on the sides and bottom. The samples were incubated statically and illuminated by one or more clip lamps with LED bulbs (Table S1) to initiate the photobiological treatment experiment. The five-gallon bucket was placed in a refrigerated incubator (Fisherbrand Isotemp$^{TM}$ BOD Refrigerated Incubator, Thermo Scientific, Waltham, MA, USA) to control the incubation temperature at a temperature other than room temperature (21–23 °C). The positions of the LED bulbs were carefully adjusted every day to ensure the consistent, desired PAR for the experiment. All the indoor experiments were conducted in duplicate. Microsoft Excel (Ver. 2404) data analysis tool and R were used for one-way analysis of variance (ANOVA) and Tukey–Kramer tests, respectively.

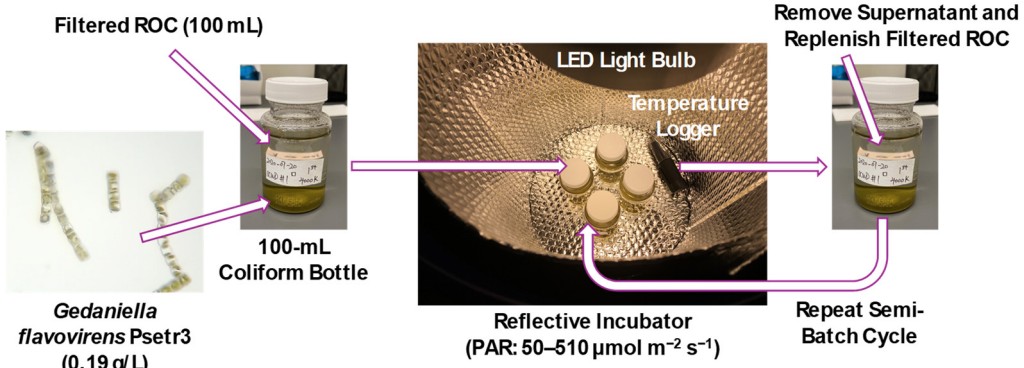

**Figure 2.** A simplified scheme of semi-batch photobiological treatment experiments. Abbreviations: ROC = reverse osmosis concentrate, LED = light-emitting diode, PAR = photosynthetically active radiation.

In this study, eight different LED light bulbs, including four bulbs with different color temperatures and four colored bulbs, were investigated (Table S1). Color temperature is widely used in the lighting industry. The four levels of color temperature were slightly different in color. The bulbs at 2700 and 3000 K are called soft white and emit mostly yellow to red lights (570–700 nm wavelengths), the 4000 K bulb is called cool white and has more blue and green lights (420–570 nm wavelengths), and the 5000 K bulb is like daylight and has strongest blue light (450 nm wavelength). Figures S1 and S2 show the light spectrum graphs of the LED bulbs used in this study.

Aliquots of supernatant samples were collected from the vessels periodically to measure selected parameters. The sample collection was performed aseptically to prevent potential contamination. Diatom biomass was periodically sampled for visual and microscopic observations to detect abnormalities such as a change in cell/colony morphology or the appearance of chloroplasts. Photomicroscopy was conducted using an AmScope T490B-DKO trinocular compound microscope equipped with an AF205 1080p HDMI C-mount microscope camera (Irvine, CA, USA). After each semi-batch cycle, the supernatant was collected by carefully decanting while keeping the diatom biomass and precipitates in the vessels.

*2.5. Photobiological Treatment—Outdoor Experiments*

Three runs of semi-batch photobiological treatment experiments were conducted outdoors on the patio area of the Roy F. Mitte Building at Texas State University in San Marcos, TX (Latitude: 29.88896, Longitude: −97.94729) on 3–22 March 2021 (Run 1), 7–10 April 2021 (Run 2), and 16 April–6 May 2021 (Run 3). Meteorological data during the experimental periods can be found in Tables S2–S4. The temperature adjacent to the vessels during the test periods was monitored using temperature data loggers, as shown in Table 2. Polycarbonate jars (500 mL) with clear acrylic (2.36 mm thickness at the beginning of Run 1) or white polypropylene lids (Run 1 after Day 5, Runs 2 and 3) were used as the reaction vessels (Figure S3). The reaction vessels were disinfected with 200 mg/L chlorine solution for 2 h and dried in a biosafety cabinet overnight. Then, 500 mL of filtered GWRS ROC was transferred into the vessels, and seed culture of *G. flavovirens* Psetr3 (0.21 ± 0.01 g/L) was added. The PAR in the mid-day (11 am–1 pm) ranged from 170 to 2100 $\mu$mol m$^{-2}$ s$^{-1}$ and from 30 to 1800 $\mu$mol m$^{-2}$ s$^{-1}$ with transparent and white lids, respectively, while the UV radiation intensities ranged from 8 to 40 W m$^{-2}$ and from 0.1 to 7.0 W m$^{-2}$, respectively (see Tables S5–S7 for more details). Similar to the indoor experiments, samples were collected from the vessels periodically for silica and NDMA analysis. Diatom biomass was also periodically sampled for visual and microscopic observations. The supernatant was collected at the end of the semi-batch cycle for additional water quality analysis.

**Table 2.** Mean temperature ($^{\circ}$C) during the outdoor experiments.

| Run | Mean | Standard Deviation | Highest | Lowest |
|---|---|---|---|---|
| 1 | 20.0 | 7.7 | 40.5 | 6.0 |
| 2 | 25.9 | 9.1 | 49.0 | 12.5 |
| 3 | 20.7 | 6.4 | 46.5 | 6.5 |

*2.6. Analytical Methods*

Table 3 summarizes the analytical methods for the water quality parameters used in this study. A Hach DR-1900 spectrophotometer (Loveland, CO, USA) was used for colorimetric analysis, while Hach Digital Titrators were used for titration analyses. A Hach IntelliCAL ISENA381 probe with a Hach HQ40d meter was used for sodium analysis. NDMA was analyzed using high-performance liquid chromatography followed by a photochemical reaction and chemiluminescence method described in Kodamatani et al. [31]. The samples were filtered through 0.45 $\mu$m polytetrafluoroethylene syringe filters upon sample collection and stored in 2.0 mL vials with closures. The method detection limit for NDMA was 0.6 ng/L.

**Table 3.** Water quality parameters tested and corresponding analytical methods.

| Parameter | Method Name | Method |
|---|---|---|
| Sodium | Ion Selective Electrode | ISENA381 |
| Calcium hardness | Titration Method with EDTA | Hach 8204 |
| Total hardness | Titration Method with EDTA | Hach 8213 |
| Iron | USEPA FerroVer® Method | Hach 8008 |
| Ammonia-N (HR) | Salicylate Method | Hach 10031 |
| Chloride | Silver Nitrate Method | Hach 8207 |
| Sulfate | USEPA SulfaVer 4 Method | Hach 8051 |
| Alkalinity | Phenolphthalein and Total Alkalinity | Hach 8203 |
| Nitrate-N (LR) | Dimethylphenol Method | Hach 10206 |
| Reactive silica | Silicomolybdate Method | Hach 8185 |
| Orthophosphate | USEPA PhosVer 3® Method | Hach 8048 |
| Chemical oxygen demand | USEPA Reactor Digestion Method | Hach 8000 |
| Chlorine, Total | USEPA DPD Method | Hach 8167 |
| Color at 455 nm | Platinum-Cobalt Standard Method | Hach 8025 |

## 3. Results

### 3.1. Impact of Light on Reactive Silica Uptake

Figure 3a shows the reactive silica uptake by *G. flavovirens* Psetr3 in GWRS ROC using LED bulbs with four different color temperatures, namely 2700, 3000, 4000, and 5000 K (Table S1). Although there were slight variations and delays in reactive silica uptake in the first semi-batch cycle, there was no marked difference in the second cycle among those temperatures. The silica uptake rate, defined as the slope of the straight portion of the uptake curve [20], was $39 \pm 2$ mg/L/day, including all the data presented in Figure 3a. Studies showed that algae generally grew better in blue and red light, since the light-harvesting pigments chlorophylls *a* and *b* are more sensitive to those colors [24,32]. However, in this study, those LED bulbs emitting stronger blue light (4000 and 5000 K) did not appear to be better than those emitting weaker blue light (2700 and 3000 K).

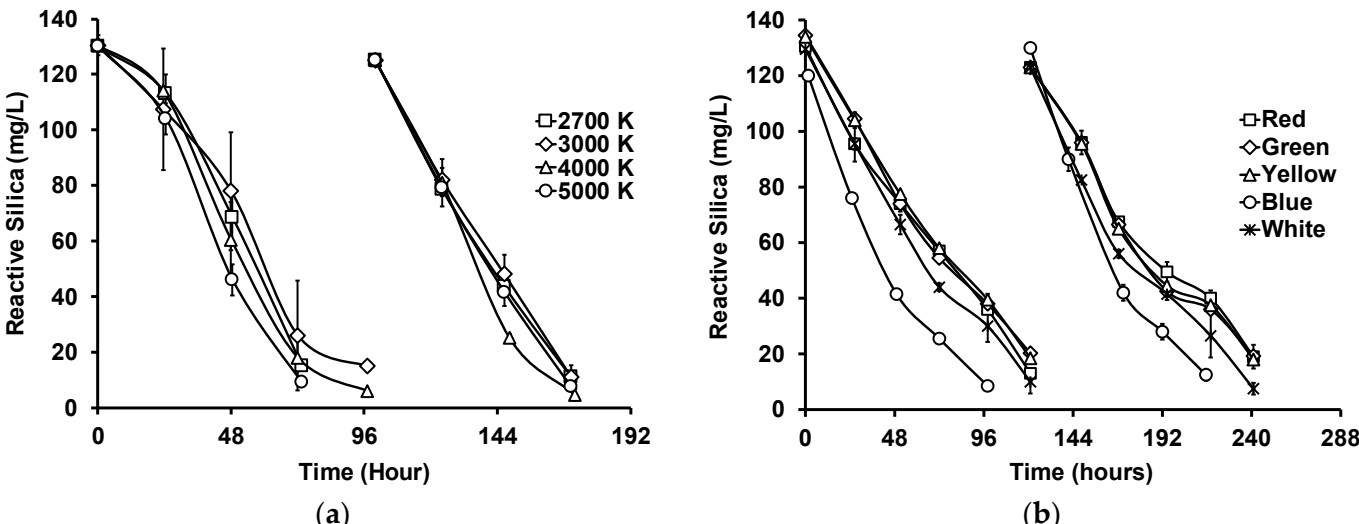

**Figure 3.** Impact of (**a**) light temperature and (**b**) color on reactive silica uptake by *G. flavovirens* Psetr3 in GWRS ROC. ((**a**) Temperature: $23.5 \pm 0.5$ °C, PAR: $200 \pm 5$ µmol m$^{-2}$ s$^{-1}$, Runtime: 172 h; (**b**) Temperature: $21 \pm 1$ °C, PAR: $50 \pm 5$ µmol m$^{-2}$ s$^{-1}$ except Green ($40 \pm 5$ µmol m$^{-2}$ s$^{-1}$), Runtime: 241 h). (Note: The error bars represent standard deviations of two replicates. Some error bars are too small to be visible).

To investigate the impact of light color on the reactive silica uptake by *G. flavovirens* Psetr3 further, LED bulbs with five distinct colors, namely red, green, yellow, blue, and white (2700 K), were evaluated (Figure 3b). In this experiment, the PAR was lowered to 40–50 µmol m$^{-2}$ s$^{-1}$ because the light output of one of the bulbs (the blue one) was very low compared with others and needed six bulbs to achieve a PAR of 50 µmol m$^{-2}$ s$^{-1}$, whereas the other colored bulbs required only one bulb. This was due to the color filter used in blue-colored bulbs, which absorbs the majority of photons associated with green to red light (500–650 nm), as shown in Figure S2. The silica uptake was slightly faster ($p = 0.002$) with blue light ($28 \pm 3$ mg/L/day) compared with the others ($22 \pm 2$ mg/L/day). However, using blue light would not be cost-effective because of the very weak light output (<15% of regular white bulbs) of blue-colored LED bulbs.

Figure 4a shows the rates of silica uptake from GWRS ROC by *G. flavovirens* as a function of the PAR ranging from 50 to 510 µmol m$^{-2}$ s$^{-1}$ using soft white LED bulbs with a color temperature of 2700 K. The silica uptake could be improved from $28 \pm 1$ mg/L/day with a PAR of 50 µmol m$^{-2}$ s$^{-1}$ to $48 \pm 7$ mg/L/day with a PAR of 310 µmol m$^{-2}$ s$^{-1}$. The improvement with PAR levels between 200 and 310 µmol m$^{-2}$ s$^{-1}$ was modest. One-way ANOVA and Tukey–Kramer tests confirmed that there was no significant difference ($p > 0.05$) in the silica uptake rates with PAR levels of 200, 310, and 510 µmol m$^{-2}$ s$^{-1}$. It

could be concluded that PAR in 200–300 µmol m$^{-2}$ s$^{-1}$ range would be optimal for the photobiological treatment of GWRS ROC using *G. flavovirens* Psetr3.

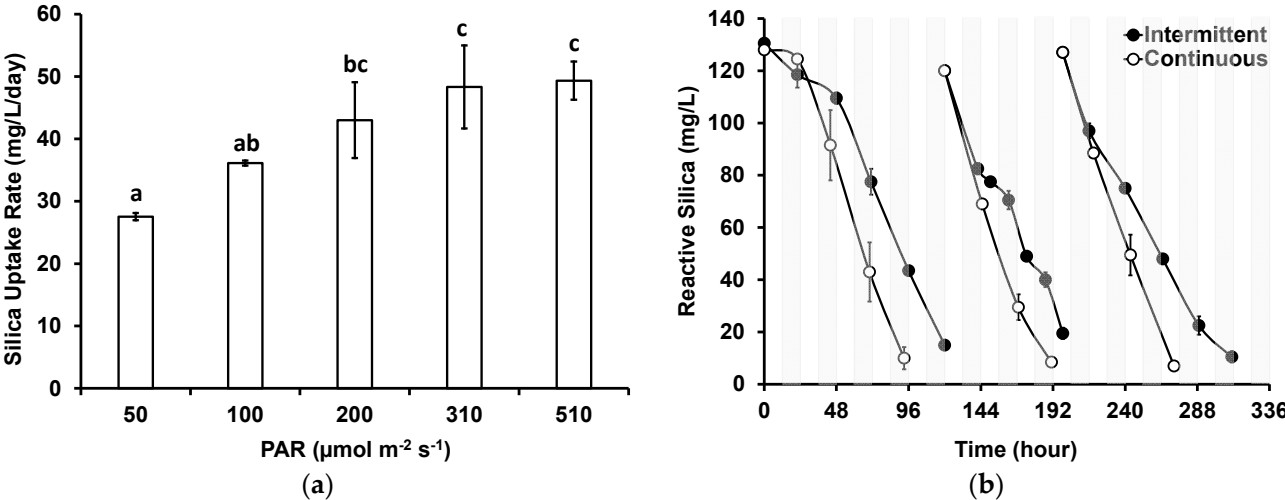

**Figure 4.** Impact of (**a**) light intensity and (**b**) intermittent light on reactive silica uptake by *G. flavovirens* Psetr3 in GWRS ROC. (Temperature: 23 ± 1 °C, 10 W LED, 2700 K, PAR: 200 ± 5 µmol m$^{-2}$ s$^{-1}$ in **b**). (Note: The error bars represent standard deviations of two replicates. Some error bars are too small to be visible. The same character above bars on (**a**) indicates no significant difference based on the Tukey–Kramer test).

The impact of intermittent light was investigated by using an on–off timer that was programmed to turn the light source on for 12 h and then turn it off for 12 h in 24 h cycles. Figure 4b shows the results, along with control groups that have received continuous light. Apparently, the reactive silica uptake was slower with intermittent light, albeit silica uptake did not completely halt during the dark periods, as seen in the second cycle where samples were collected during the dark periods. This is probably because the photosynthesis and cell division, for which silica uptake is required, are independent in diatoms [33]. A similar observation was noted when another strain of *G. flavovirens* (*P. trainorii* PEWL001) was evaluated in an outdoor PBR treating GWRS ROC [18]. The silica uptake rates for the intermittent and continuous light groups were 27 ± 2 and 41 ± 3 mg/L/day, respectively.

### 3.2. Impact of Temperature on Reactive Silica Uptake

In addition to the ambient temperature (21–23 °C), the photobiological treatment of GWRS ROC at three additional temperatures at 10, 30, and 40 °C was evaluated (Figure 5). While there was no apparent difference in the reactive silica uptake rates between 23 and 30 °C, it was slower at 10 °C. Further, there was a prolonged lag period (three days) for the diatoms to acclimatize to the lower temperature. At 40 °C, no reactive silica uptake occurred, but a slight increase in concentration (up to 150 mg/L) was observed. The biomass lost dark green pigments upon incubation at 40 °C. The microscopic analysis confirmed that *G. flavovirens* Psetr3 cells were bleached (Figure 6a), as compared with the healthy cells grown at 23 °C (Figure 6b). It can be concluded that the optimum temperature of the photobiological treatment is around 23–30 °C.

### 3.3. Sunlight Experiments

Three runs of the photobiological treatment of GWRS ROC were conducted using natural sunlight as a light source. In Run 1, no reactive silica uptake was observed in the first five days (Figure 7a), although the color of the biomass was still dark green based on visual observation. Instead, the reactive silica concentration increased by 15 mg/L, which indicated the death of diatoms. Microscopic analysis showed some dispersed bleached diatom cells, but there were still cells with visible chloroplasts. In addition, there were

some large aggregates (20–40 μm in diameter) of diatom frustules (Figure 8a). In this run, lids fabricated with transparent acrylic sheets were used to maximize the PAR input to the vessel. However, it was found that the UV transmittance of the transparent acrylic sheet was approximately 90% and the UV radiation was as high as to 7 W m$^{-2}$. We suspected that this UV radiation inactivated *G. flavovirens* Psetr3 cells on the first day of Run 1.

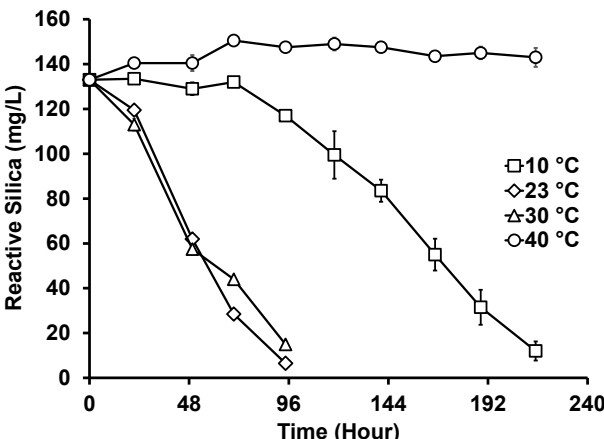

**Figure 5.** Reactive silica uptake by *G. flavovirens* Psetr3 in GWRS ROC at various temperatures. (10 W LED, 2700 K, PAR: 200 $\pm$ 5 μmol m$^{-2}$ s$^{-1}$, Runtime: 215 h) (Note: The error bars represent standard deviations of two replicates. Some error bars are too small to be visible).

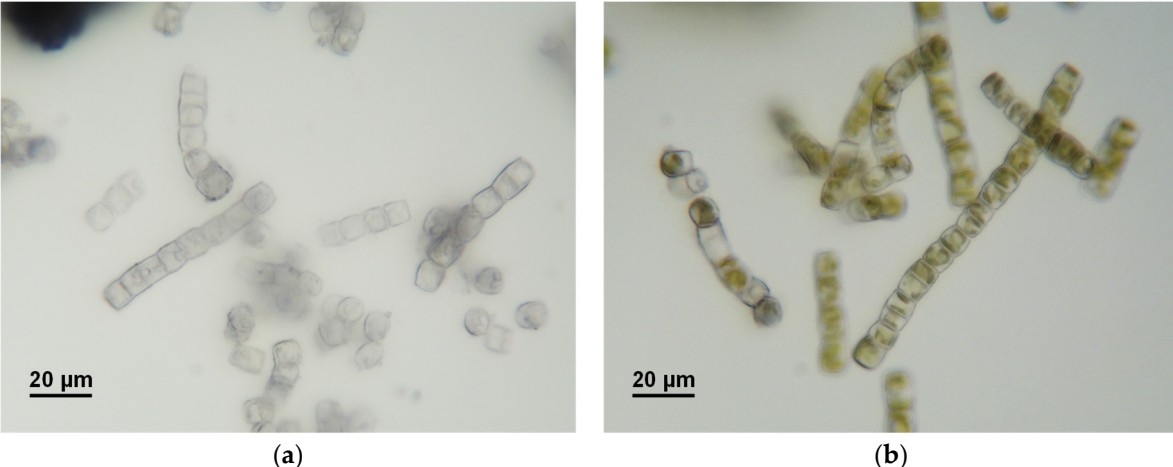

(**a**)　　　　　　　　　　　　　　　　　　　(**b**)

**Figure 6.** Photomicrographs of *G. flavovirens* Psetr3 incubated at (**a**) 40 and (**b**) 23 °C in GWRS ROC for three days (96 h in Figure 5).

On Day 5, the addition of more biomass and the replacement of the acrylic lids with the original white polypropylene lids resulted in active silica uptake. Thus, in the subsequent runs, white polypropylene lids that block UV light were used. See Figure S4 for the typical PAR and UV profiles during a day with or without a white lid.

In Run 2, the reaction vessels were moved to an area on the patio with less direct sunlight. However, no reactive silica uptake was observed for four days in this run either (Figure 7a). This time, the high temperature during this period was suspected to be the cause. The temperature measured adjacent to the reaction vessels (Figure S5b) showed the diatoms were exposed to >40 °C for more than 2 h on Days 1 and 2 due to the heat transferred from the patio floor, which apparently killed all the *G. flavovirens* Psetr3 cells (Figure 8b). This confirmed the vulnerability of this diatom towards high temperatures, as observed in Figure 5.

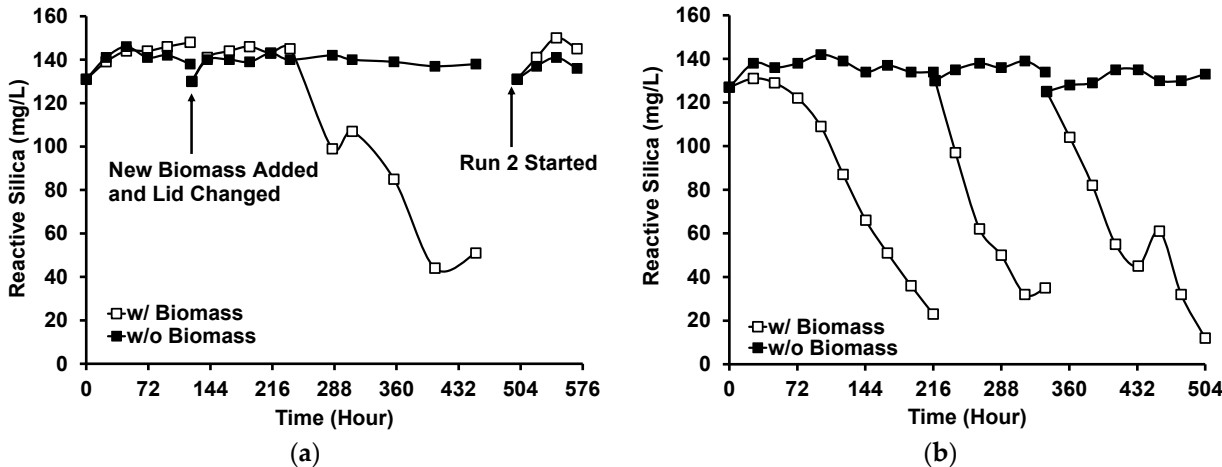

**Figure 7.** Reactive silica uptake by *G. flavovirens* Psetr3 in GWRS ROC using sunlight as a light source: (**a**) Runs 1 and 2 and (**b**) Run 3. (Temperature: 6–45 °C, PAR: Up to 1800 μmol m$^{-2}$ s$^{-1}$, Runtime: Run 1: 452 h; Run 2: 70 h; Run 3: 504 h).

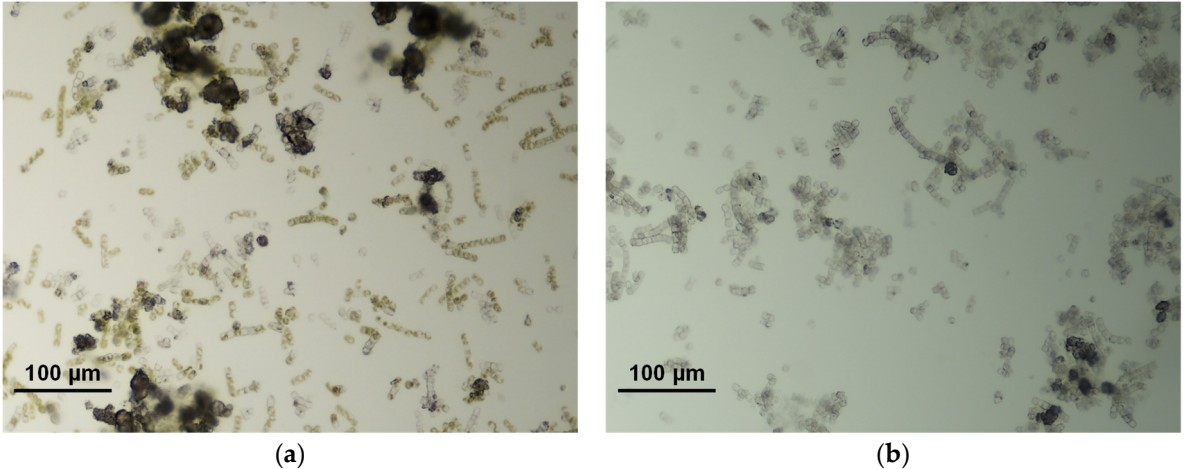

**Figure 8.** Photomicrographs of *G. flavovirens* Psetr3 exposed to (**a**) direct sunlight in the outdoor experiment Run 1 (after 120 h) and (**b**) high temperature in Run 2 (after 70 h).

Run 3 was conducted in the subsequent weeks. The temperature was relatively low during that time (Table 2). The reaction vessels were moved into a more shaded area to eliminate direct sunlight and the warm patio floor. A visible light emission spectrum from the experimental location at midday is shown in Figure S6. In this run, the photobiological treatment was successful (Figure 7b). No reactive silica removal occurred in the control vessel where no *G. flavovirens* Psetr3 biomass was added (i.e., closed squares in Figure 7b). The temperature profile data (Figure S5c) showed that brief spikes in temperature > 40 °C occasionally occurred in the second and third cycles. However, those incidents did not inactivate the diatoms as they lasted <10 min.

In Run 3, the reactive silica uptake was slower (17 mg/L/day) in the first cycle compared with the second cycle (27 mg/L/day). This was presumably due to the lower temperatures (daily average: 15.9 ± 2.9 °C, daily low: 10.4 ± 4.6 °C) in the first cycle than those in the second cycle (daily average: 22.4 ± 1.7 °C, daily low: 17.8 ± 5.2 °C) (Table S6). The reactive silica uptake slowed in the third cycle (15 mg/L/day), possibly due to the high temperature (Figure S5c) and/or high UV irradiation (0.9 W m$^{-2}$, Table S7).

Figure 9 shows the removal of NDMA from GWRS ROC during Run 3 of the outdoor experiment with and without *G. flavovirens* Psetr3. The degradation of NDMA was slower in the vessel with *G. flavovirens* biomass. This suggests that the mechanism of NDMA

degradation is primarily governed by direct photolysis and that the presence of biomass scavenged the photons available for NDMA photolysis. No degradation of NDMA was observed when LED light bulbs were used as a light source. This result contradicted our previous study, where significant NDMA removal (67%) was observed in LED-based photobiological treatment [18]. The previous result could have been an artifact or in error because LED light bulbs do not emit light at <300 nm (Figure S2), the wavelength absorbed by NDMA [28]. The photodegradation of *N*-nitrosamines (including NDMA) by natural sunlight is well known [34,35].

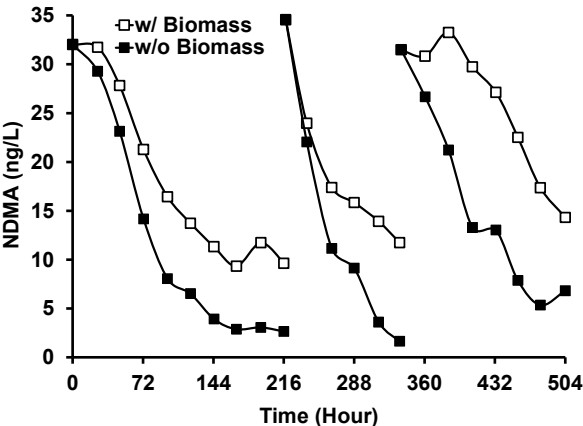

**Figure 9.** Removal of *N*-nitrosodimethylamine (NDMA) during the photobiological treatment of GWRS ROC with *G. flavovirens* Psetr3 using sunlight as a light source (Run 3). (Temperature: 6–45 °C, PAR: 13–606 µmol m$^{-2}$ s$^{-1}$, Runtime: 504 h).

## 4. Discussion

Light spectral selection has been employed to optimize and improve the growth of microalgae and lipid production in PBRs [24,32,36,37]. Warmer colors like red and yellow lights are within 550–700 nm, while green and blue are <500 nm in wavelength (Figure S1). Those studies reported blue (425–450 nm) and red (600–700 nm) ranges are generally effective for algal photosynthesis. However, in this study, there was only a minor difference in the silica uptake rates among these colors (blue: 29 ± 3 mg/L/day, others: 22 ± 2 mg/L/day) in *G. flavovirens* Psetr3. This was possibly due to the yellow background color (145 PtCo unit) of the GWRS ROC, which could filter some light at shorter wavelengths (Figure S7). There would be no practical benefit of light spectral selection in this photobiological treatment process. Therefore, soft white LED bulbs with a color temperature of 2700 K, which are the least expensive and readily available, can be used in future work.

Shi et al. [21] reported that the growth and silica uptake could be accelerated with higher PAR (80 µmol m$^{-2}$ s$^{-1}$) in freshwater diatoms such as *Cyclotella meneghiniana*, *Stephanodiscus parvus*, and *Synedra acus* compared with lower PAR (25 µmol m$^{-2}$ s$^{-1}$), although the growth of some strains that were adapted to low light intensity could be inhibited if the PAR was above 80 µmol m$^{-2}$ s$^{-1}$. In this study, higher PAR levels (200–300 µmol m$^{-2}$ s$^{-1}$) were found to be optimum (Figure 4a). These levels are much lower than the PAR of full sunlight during the daytime. The result of intermittent light study (Figure 4b) showed that continuous illumination would not be required. These findings indicate potential operation of the diatom PBRs using sunlight as a light source.

The impact of temperature on the growth of diatoms and other microalgae is well known [27,38]. For example, Li and Dickie [39] showed that the carbon and hydrogen uptakes in marine microalgae were temperature dependent and that their optima were between 17 and 27 °C. Mitrovic et al. [40] also reported that the growth of the freshwater diatom *Cyclotella meneghiniana* could slow down at temperatures below 10 °C and above 30 °C. These observations are in line with the findings of this study using *G. flavovirens* Psetr3.

The impacts of light and temperature observed in the laboratory experiments under a controlled environment were verified in the sunlight experiments where those conditions become more variable and unpredictable. The high temperature (>40 °C) during Run 2 killed the brackish diatoms, while lower temperatures (10–15 °C) slowed down the silica uptake in Run 3. The detrimental impact of direct sunlight and UV radiation was realized during the sunlight experiments, which is consistent with our previous studies using a different strain of *G. flavovirens* in Southern California [13,16]. The inhibition of photosynthesis by UV radiation in the marine diatom *Thalassiosira pseudonana* is also known [35]. The observed silica uptake during the outdoor experiments was comparable to our previous study, although the maximum reactive silica uptake rate was approximately 30–40% lower than in the indoor experiments, as well as another outdoor diatom PBR study in Southern California [18]. This was probably due to the variable and harsher climate in Central Texas (6–49 °C; Table 2) compared with milder Southern California climate (18–40 °C) during the experiments. Fluctuating temperatures, both high and low ones, along with UV radiation, represent potential major risk factors for the seamless continuous operation of the diatom PBR-RO process and must be effectively mitigated. Disruption or poor performance of the PBR due to suboptimal temperature and UV exposure could impact the secondary RO system, resulting in reduced freshwater recovery.

While no synergistic effect of diatom photobiological treatment on NDMA removal was observed compared to simple solar treatment, the significant NDMA removal demonstrated in this research (Figure 9) remains advantageous for the implementation of photobioreactors at AWPFs. In regions like California, where nitrosamine concentrations in purified water are regulated for indirect and direct potable water reuse projects [1,30], effective NDMA removal is crucial.

Additional research would be required to further improve the silica uptake in outdoor PBRs, since the use of sunlight is ultimately desired to develop a cost-effective diatom PBR-RO process. It is also desirable to search for diatom strains, including mixed cultures, with better UV and/or temperature resistance to construct more robust PBR systems. These follow-up research works are currently underway.

## 5. Conclusions

This study investigated the optimal light and temperature conditions for maximizing silica uptake by the brackish diatom *G. flavovirens* Psetr3 in ROC from OCWD GWRS AWPF. While color temperature (2700–5000 K) had no significant impact on silica uptake ($39 \pm 2$ mg/L/day) at a PAR level of 200 $\mu$mol m$^{-2}$ s$^{-1}$, blue-colored LED bulbs slightly improved uptake (+22%) despite impractical light output. Increasing the PAR to 200–300 $\mu$mol m$^{-2}$ s$^{-1}$ enhanced silica uptake, while intermittent light slowed uptake by 34% compared to continuous illumination. The optimum temperature ranged from 23 to 30 °C, with uptake slowing at 10 °C and a loss of diatom viability at 40 °C. Outdoor experiments confirmed the impact of light and temperature on silica uptake, with the high temperatures (>40 °C) and UV radiation from sunlight negatively affecting uptake. Outdoor uptake rates (15–27 mg/L/day) were generally lower than indoor rates (40–50 mg/L/day), primarily due to fluctuating temperatures. These findings underscore the importance of light management and temperature control in the diatom PBR-RO process using sunlight. Sunlight-based photobiological treatment effectively removed NDMA from AWPF ROC, likely through direct photolysis rather than biological degradation. This highlights the potential of a sunlight-based process for ROC treatment, especially given NDMA's significance in potable reuse projects. These results inform ongoing pilot testing and full-scale lifecycle cost analysis, demonstrating the technical and economic feasibility of this green technology. Ultimately, the diatom PBR-RO process promises to enhance freshwater recovery from reclaimed and brackish water, supporting sustainable desalination and potable water reuse projects in arid and semi-arid regions.

**Supplementary Materials:** The following supporting information can be downloaded at: https://www.mdpi.com/article/10.3390/su16104052/s1, Figure S1: Light emission spectra of the four white LED bulbs tested in this study, Figure S2: Light emission spectra of the five colored LED bulbs tested in this study, Figure S3: Experimental setup—Outdoor experiments, Figure S4: Hourly (a) PAR and (b) UV measurements in the 500 mL polycarbonate jars used in the outdoor experiments on May 14, 2021, Figure S5: Temperature profiles during the outdoor experiment: (a) Run 1, (b) Run 2, and (c) Run 3, Figure S6: Typical light emission spectrum at the outdoor experimental location in the mid-day (at 2 pm), Figure S7: UV-vis absorbance spectrum of GWRS ROC, Table S1: LED bulbs used in this study, Table S2: Meteorological data during the outdoor experiment (Run 1), Table S3: Meteorological data during the outdoor experiment (Run 2), Table S4: Meteorological data during the outdoor experiment (Run 3), Table S5: Daily PAR and UV measurements during the outdoor experiment (Run 1), Table S6: Daily PAR and UV measurements during the outdoor experiment (Run 2), Table S7: Daily PAR and UV measurements during the outdoor experiment (Run 3).

**Author Contributions:** Conceptualization, H.G. and K.I.; methodology, H.G., H.K. and K.I.; software, H.G., S.S. and H.K.; formal analysis, H.G., S.S., H.K. and K.I.; investigation, H.G. and K.I.; resources, S.S., H.K., T.F., K.P.I. and K.I.; data curation, H.G. and K.I.; writing—original draft preparation, H.G. and K.I.; writing—review and editing, H.G., S.S., H.K., T.F., K.P.I. and K.I.; visualization, H.G. and K.I.; supervision, K.I.; project administration, K.I.; funding acquisition, K.I. All authors have read and agreed to the published version of the manuscript.

**Funding:** The materials presented in this paper are based upon work supported by the United States Bureau of Reclamation Desalination and Water Purification Research Program (Award #: R21AC10106, Awarded to K.I.).

**Institutional Review Board Statement:** Not applicable.

**Informed Consent Statement:** Not applicable.

**Data Availability Statement:** Supplementary data for this article can be found at (data provided as Supplementary Materials).

**Acknowledgments:** The authors would like to thank Jacob A. Palmer, Lokendra Acharya, Emma M. Clow, Dennis Davilla, Saul Gozalez, and Dustin M. Walker at Texas State University, San Marcos, TX for their technical assistance.

**Conflicts of Interest:** The authors declare that they have no known conflicts of interests or personal relationships that could have appeared to affect the work reported in this paper.

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
