# Peer review of "Optimization of Dissolved Silica Removal from Reverse Osmosis Concentrate by Gedaniella flavovirens for Enhanced Water Recovery"

_sustainability, doi:10.3390/su16104052_

Round 1

Reviewer 1 Report

Comments and Suggestions for Authors

The research aims to determine the optimum conditions for the photobiological treatment of ROC obtained from a full-scale advanced water purification facility using Gedaniella flavovirens Psetr3.

The work is very well presented. The objectives and methods are clear. The discussion is good and the abstract and conclusions are in line with the results and their analysis.

In the introduction it is necessary to better explain the advantages of RO in relation to other methods such as UF, NF, disinfection, electroflotation, ozonation or electrochemical methods. It would be worth explaining.

1) Section 2.4

It would be much clearer if a figure with the semi-batch photobiological treatment experiments is presented.

Figures S1 and S4 should be part of the text.

Comments on the Quality of English Language

None.

Author Response

We are grateful for the reviewer's positive feedback on our manuscript. We have taken their comments into careful consideration and made revisions accordingly. 

Reviewer 2 Report

Comments and Suggestions for Authors

The article titled “Optimization of Dissolved Silica Removal from Reverse Osmosis Concentrate by Gedaniella flavovirens for Enhanced Water Recovery” represents the treatment of reverse osmosis concentrate (ROC) using photobiological treatment (brackish dia-toms).

1-  I found that the concluding remark “The results of this study suggest the importance of light and temperature management in this photobiological process” in Lines 23-25 is very general and well-known; authors are encouraged to show the novelty of their study!!

2-  The study objectives in Lines 70-81 are not clear

3-  In Table 1, the authors are requested to additional column for the water quality standard for comparison

4-  For Lines 133-134; add a separate section for this ANOVA statistical method

5-  What are the devices used to measure the water quality parameters in Table 3

6-  What are the biochemical reactions used to describe the effect of light in section “3.1. Impact of Light”

7-  At what operational time the images in Fig. 5 were captured.

8-  What are the biochemical reactions used to describe the effect of temperature in section “3.2. Impact of Temperature”

9-  At what operational time the images in Fig. 7 were captured.

10-  In section “4. Discussion” the authors are requested to draw a schematic diagram illustrating the effects of light and temperature on RO performance

11-  The Conclusion section is very lengthy!!

Comments on the Quality of English Language

Moderate editing of English language required

Author Response

We appreciate the thorough review provided by the reviewer. Their insightful comments have provided valuable perspectives for improving our manuscript. We have carefully considered each point raised and have made revisions accordingly. While we may not agree entirely with all the comments, we acknowledge their importance in strengthening the quality of our work. Thank you for your time and constructive feedback.  Please see the attachment.

Reviewer 3 Report

Comments and Suggestions for Authors

This well-written manuscript provides adequate background and support information. However, including the suggested improvements can further enhance its conciseness and impact.

1.     In Figure 1: "Simplified scheme of the diatom photobioreactor (PBR)," please identify the full names of MF, RO, UF, AOP, etc. presented in the figure.

2.     The authors should include the standard deviation of the analysis results for each parameter in Table 1.

3.     For Figures 5 and 7, which are photomicrographs, the authors should describe the method and technique used for the analysis.

4.     Figures 2, 4, 6, and 8 have a y-axis labeled as time (h). To improve clarity for the reader, the authors should include the run time in the title of each figure. (Improved flow).

5.     In the discussion section, the authors should consider including more relevant information for comparing the obtained results.

6.     Line 374-377: For a stronger statement about the 22% improvement in silica uptake with blue LED light, consider adding the statistical significance (p-value).

7.     The conclusion (page 11, sentences 379-380) seems to contain redundant information. Consider revising these sentences to improve clarity.

Comments on the Quality of English Language

The authors should have the manuscript proofread to enhance the clarity and readability of the information provided in the text.

Author Response

We appreciate the thorough review provided by the reviewer. Their insightful comments have provided valuable perspectives for improving our manuscript. We have carefully considered each point raised and have made revisions accordingly. Thank you for your time and constructive feedback. Please see the attachment.

Round 2

Reviewer 1 Report

Comments and Suggestions for Authors

The authors significantly improved the quality of the manuscript.

Comments on the Quality of English Language

None

Author Response

We sincerely thank the reviewer for generously dedicating their time and expertise to review our manuscript.  We are happy to hear that the revised version is satisfactory.  Thank you very much!

Reviewer 2 Report

Comments and Suggestions for Authors

The Conclusion section should be outlined in a one paragraph

Comments on the Quality of English Language

Minor editing of English language required

Author Response

We sincerely thank the reviewer for generously dedicating their time and expertise to review our manuscript.  As suggested, the paragraphs in the Conclusions section have been combined into one paragraph.  Thank you again for your valuable feedback.